# Clinical Presentations, Pathogenesis, and Therapy of Sarcoidosis: State of the Art

**DOI:** 10.3390/jcm9082363

**Published:** 2020-07-24

**Authors:** Francesca Polverino, Elisabetta Balestro, Paolo Spagnolo

**Affiliations:** 1Asthma and Airway Disease Research Center, University of Arizona, Tucson, AZ 85718, USA; 2University Hospital of Padua, 35100 Padua, Italy; elisabetta.balestro@aopd.veneto.it (E.B.); paolo.spagnolo@unipd.it (P.S.)

**Keywords:** sarcoidosis, pathogenesis, treatment, antigen, immunity, systemic involvement, clinical presentation

## Abstract

Sarcoidosis is a systemic disease of unknown etiology characterized by the presence of noncaseating granulomas that can occur in any organ, most commonly the lungs. Early and accurate diagnosis of sarcoidosis remains challenging because initial presentations may vary, many patients are asymptomatic, and there is no single reliable diagnostic test. Prognosis is variable and depends on epidemiologic factors, mode of onset, initial clinical course, and specific organ involvement. From a pathobiological standpoint, sarcoidosis represents an immune paradox, where an excessive spread of both the innate and the adaptive immune arms of the immune system is accompanied by a state of partial immune anergy. For all these reasons, the optimal treatment for sarcoidosis remains unclear, with corticosteroid therapy being the current gold standard for those patients with significantly symptomatic or progressive pulmonary disease or serious extrapulmonary disease. This review is a state of the art of clinical presentations and immunological features of sarcoidosis, and the current therapeutic approaches used to treat the disease.

## 1. Definition and Clinical Presentations

Sarcoidosis is a systemic granulomatous disorder that remains an enigma and challenges both researchers and clinicians due to its unknown cause, heterogeneous clinical presentation, and unpredictable outcome, occasionally severe and even lethal, often with no therapy able to change the course of the disease [1]. This huge variability led scientists to search for clinical patterns of phenotypic variables in order to help the diagnosis and avoid, when possible, invasive procedures. Sarcoidosis can affect individuals of all ages but most commonly affects men aged 30 to 50 years, and women aged 40 to 60 years [2]. Phenotypic differences in sarcoidosis subsets may relate, at least partially, to the variability of the organs involved.

### 1.1. Lung

The frequency of organ involvement at the time of diagnosis is summarized in Table 1 [3,4]. The lung is the most commonly affected organ, although with a frequency that varies according to race, sex, and age. Thoracic involvement occurs in 90% of patients with bilateral symmetrical hilar adenopathy representing the most common thoracic manifestation (Figure 1), whereas unilateral hilar adenopathy occurs in only 3% to 5% of patients. 

### 1.2. Skin 

Even though the thorax is the most common site of disease, skin involvement occurs in at least 25% of patients and is often overlooked. Indeed, cutaneous sarcoidosis is often attributed to other causes, mainly due to its wide range of manifestations, which include erythema nodosum, maculopapular lesions, hyper- and hypo pigmentated areas, keloid formation, and subcutaneous nodules [5]. With the exception of erythema nodosum, which consists of a panniculitis and does not contain granulomas, skin lesions are generally florid areas of granulomatous inflammation, and the diagnosis of sarcoidosis can be readily made by skin biopsy. Erythema nodosum may accompany acute presentations of the disease, is more common in women, and tends to be associated with a favorable prognosis, while manifestations such as lupus pernio [6], often described in African Americans and Puerto Ricans, are associated with chronic disease and worse prognosis. 

### 1.3. Eye

Eye involvement may precede the diagnosis of sarcoidosis by years or even decades and occurs in up to 25% of patients depending on race [7]. For example, only 30% of patients in the United States have sarcoidosis-related eye illness vs. more than 70% in Japan. The most common manifestations are anterior uveitis with acute pain and blurred vision, while photophobia is reported in fewer cases [8]. 

### 1.4. Heart 

Cardiac involvement in sarcoidosis is much more common than clinically appreciated, and it can be present without the involvement of other organs. The three major manifestations of cardiac sarcoidosis are: conduction abnormalities, ventricular arrhythmias, and congestive heart failure. Granulomatous infiltration of the myocardium is responsible for arrhythmias and cardiomyopathy [9]. In particular, the atrioventricular (AV) block is the most frequent type of arrhythmia, whereas ventricular tachycardia and supraventricular arrhythmia are less common. Chronic heart failure may occur both with reduced ejection fraction from dilated cardiomyopathy and with preserved ejection fraction from restrictive cardiomyopathy [10,11]. Since cardiac sarcoidosis can occur in the absence of apparent disease elsewhere, it should be suspected and investigated in any non-ischemic form of cardiomyopathy, particularly when arrhythmias are prominent [12,13]. Furthermore, cardiac involvement can be observed in patients with advanced pulmonary fibrosis due to Pulmonary Arterial Hypertension (PAH) [14]. 

### 1.5. Brain

Neurosarcoidosis is reported in 5–10% of sarcoidosis patients and appears to be equally frequent across ethnic groups. Any part of the nervous system can be affected, and among all parts of the central nervous system (CNS), cranial neuropathy and meningeal involvement are the most common manifestations. Furthermore, II, VII, and VIII cranial nerves are the most commonly involved, even if all cranial nerves have been reported to be potentially affected. Meningitis accounts for 10% to 20% of cases of neurosarcoidosis, although the reported frequency of subclinical leptomeningeal damage is much higher [15]. Finally, brain parenchymal disease is less common, and patients may present with seizure, headache, or cognitive/behavioral problems. Likewise, spinal cord disease and peripheral neuropathy are quite rarely reported [16,17,18].

## 2. Subtypes of Sarcoidosis and Outcomes 

The clinical presentation of sarcoidosis is highly heterogeneous, and the onset can be acute or subacute/chronic. It has been hypothesized that different immunologic features underlie these diverse presentations. From both clinical and immunological standpoints, sarcoidosis is mainly classified in Löfgren’s syndrome (LS) and non-Löfgren’s syndrome. In more than half of the cases of LS, the disease resolves spontaneously within 2 years, on average [19,20]. After five years, recovery is much less likely. In terms of clinical development and outcome, sarcoidosis can be classified into acute (≤2 years) and chronic (≥3–5 years). 

### 2.1. Lofgren’s Syndrome (LS)

The typical combination of symptoms with acute onset such as fever, erythema nodosum, and arthritis accompanied by bilateral hilar lymphadenopathy at chest X-ray is defined as LS (Figure 1) [21]. LS is highly associated with the human leukocyte antigen (HLA) B8 (HLA-B*) serotype [22,23]. Further, a strong association between HLA-B8/DR3 and acute onset of symptoms has been shown in patients with sarcoidosis with bilateral hilar lymphadenopathy, suggesting the key role of immunogenetics in the multifaceted disease presentation [24]. Rapid onset of acute disease is generally associated with spontaneous resolution and excellent prognosis. LS associated with the HLA-DQB1*0201 allele is associated with good prognosis in European patients, whereas HLA-DRB1*0301 has been associated with an equally favorable prognosis in the Swedish population, even without therapy [25,26]. In the great majority of cases, patients with LS recover from symptoms and/or radiologic abnormalities, either spontaneously or after corticosteroid treatment. The Heerfordt-Waldenström syndrome is another rare form of sarcoidosis with acute presentation and typical signs and symptoms such as parotitis, facial palsy, anterior uveitis, and fever. The radiographic appearance, although nonspecific, shows a typical enlargement of the parotid gland(s) and enlargement of cervical lymph nodes. 

### 2.2. Subacute/Chronic Sarcoidosis 

The non-acute form, or non-LS sarcoidosis, is much more heterogeneous than LS and presents with a wide range of nonspecific symptoms and with an insidious/subacute onset [27]. Subacute to chronic presentation often include cough, shortness of breath, arthralgia, fatigue, chest pain, muscle pain, night sweats, and weight loss. Fatigue is a very common complaint (up to 50–70% of the patients) and leads to impaired quality of life and disability [28]. Chronic sarcoidosis is associated with an increased risk of developing fibrosis (pulmonary and extrapulmonary), pulmonary arterial hypertension (PAH), and other persistent disabling symptoms, as well as permanent loss of lung function associated with impaired quality of life [3]. Extrapulmonary manifestations associated with unfavorable prognosis include lupus pernio, chronic uveitis, chronic hypercalcemia, nephrocalcinosis, cystic bone lesions, and myocardial involvement. 

Progression to irreversible fibrosis. Pulmonary fibrosis (PF) represents a rare (20% of chronic forms) yet poorly studied phenotype of sarcoidosis (Figure 2). Chronic granulomatous inflammation of the lung may lead to the development of pulmonary fibrosis, which is associated with an increased risk of mortality. Pulmonary fibrosis may manifest as either progressive disease or an exhausted and quiescent process with inactive granulomatous inflammation. Interestingly, a local shift from Th1 to Th2 responses has been associated with progression from granulomatous inflammation to fibrotic disease [29,30]. However, it remains to be clarified whether sarcoidosis-associated PF: (1) is a phenotype that begins early in the course of the disease or whether it develops as a response to an excessive immune response; and (2) once initiated, can progress independently of active granulomatous inflammation. PAH occurs in at least 5% of patients with sarcoidosis, mainly affected by the chronic form. PAH is the most frequent complication of advanced PF and is associated with poor prognosis. A number of mechanisms may underlie the development of PAH, including direct vascular involvement by the granulomatous process, PF, extrinsic arterial compression by lymphadenopathies, or pulmonary veno-occlusive disease [31]. The majority of patients with end-stage PF show PAH, with a prevalence that is significantly higher than that reported for other fibrotic interstitial lung diseases (Figure 3) [32]. PF with PAH is generally complicated by the development of respiratory failure and infections such as mycetoma [33]. 

### 2.3. Mortality in Sarcoidosis

Patients with sarcoidosis have a lower survival rate than the general population [34], with a mortality rate reported to be up to 7.6% [35]. In a study conducted to evaluate mortality rates among French decedents with sarcoidosis from 2002 to 2011, the mean age at death was lower than the general population by about six years [36]. In addition, in a Swedish cohort, individuals with incident sarcoidosis (2003–2013) had a higher risk of death compared to the general population [37]. Sarcoidosis was often associated with death because of respiratory failure due to advanced PF, cardiovascular diseases, and neurologic disease with a different rate depending on the country of origin. For example, in western countries, most fatalities were due to end-stage lung disease, whereas in Japan, the main cause of death was heart failure [38]. Less common causes of mortality include hemoptysis from mycetoma and lymphoma [35].

## 3. Pathogenesis of Sarcoidosis

Sarcoidosis is characterized by non-infectious non-caseating granulomas, comprised primarily of macrophages that differentiate to epithelioid cells, which subsequently fuse to form multinucleated giant cells. CD4+ T helper cells, the specificity of which is unknown, are interspersed in the granuloma while CD8+ T cells, regulatory T cells (Tregs), fibroblasts, and B cells surround the periphery [39]. This heterogeneous cell population suggests that both innate and adaptive immune responses contribute to the onset and the progression of the disease. To date, however, the pathogenesis of sarcoidosis represents an unresolved immunological paradox. In sarcoidosis, affected organs present an intense immune response, yet at the same time, a state of immune anergy is established, as indicated by a reduced delayed-type hypersensitivity to tuberculin and common antigens (Figure 4) [40]. 

### 3.1. Putative Antigens 

The involvement of both the innate and the adaptive immune cells calls into question the nature of the antigen/s involved in the pathogenesis of sarcoidosis. Thus far, apart from the tuberculin known to trigger the formation of the sarcoid granuloma, many have been the candidate antigens associated with the onset and the progression of the disease. Bacterial DNA has been identified in sarcoidosis lesions [41] as well as autoantigens belonging to the major histocompatibility complex (MHC) class II molecules on antigen-presenting cells, recognized by the T-cell receptor (TCR) of the responding T-cells of sarcoidosis patients and leading to their clonal expansion [42]. Strong T cell responses to vimentin, a peptide derived from the cytoskeleton, have been found in a subset of patients with sarcoidosis with a specific HLA type [42]. Similarly, stimulation of peripheral blood mononuclear cells (PBMCs) from patients with sarcoidosis with vimentin triggers increased secretion of cytokines able to sustain the immune response [43], suggesting a self-perpetuating mechanism similar to that known to occur in autoimmune diseases.

Overall, the immune cascade triggered by the antigen/s is much better known compared to the nature of the antigen itself. We here describe the role of innate and adaptive immune responses in the onset and the progression of sarcoidosis. 

### 3.2. Innate Immune Cells 

Alveolar macrophages (AMs) are involved both in innate and adaptive immune responses in sarcoidosis [44]. AMs produce a range of pro-inflammatory cytokines, such as tumor necrosis factor-α (TNF-α), which drives granuloma formation in sarcoidosis [45]. In addition, AMs function as antigen-presenting cells (APCs), interacting with T-cells via human leukocyte antigen (HLA) molecules and T-cell receptors [46]. Activation of the mammalian target of rapamycin complex 1 (mTORC1) pathway in macrophages promotes excessive granuloma formation in mice and is associated with macrophage proliferation and disease progression in patients with sarcoidosis [47]. An analysis of peripheral blood monocytes from sarcoidosis patients found enrichment of activated monocytes available to populate sites of granulomatous inflammation and a higher prevalence of cells expressing CD11c, possibly representing a subpopulation of monocyte-derived cells that could differentiate to dendritic cells, thus linking innate and adaptive responses in sarcoidosis [48]. There is an increased number of phagocytic monocytes in the blood of sarcoidosis patients to control subjects. Similarly, circulating monocytes also display higher Toll Like Receptor TLR2 and TLR4 expression, inducing Th1 and Th2 responses, respectively [46]. Bronchoalveolar lavage (BAL) cells from patients with sarcoidosis produce more TNF-α and IL-6 in response to TLR2 agonists compared to healthy controls, while PBMCs from sarcoidosis patients have impaired TLR2 responses [49]. Importantly, macrophage activation status and polarization exert a dominant effect on the outcome of granulomatous inflammation. Granulomatous inflammation in pulmonary sarcoidosis has historically been linked to a compartmentalized elevation of T-helper cell type 1 (Th1) cytokines, including but not limited to IL-2, IFN-γ, and TNF-α. This cytokine milieu is believed to drive classical pro-inflammatory (M1) macrophage activation, whereas the proportion of anti-inflammatory M2 macrophages tends to be higher in other types of interstitial lung disease, including idiopathic pulmonary fibrosis [50]. However, increases in M2 macrophages have been recently reported in granulomas of patients with sarcoidosis as compared with tuberculous granulomas [51]. Interestingly, M2 macrophages are capable of differentiating into fibrocyte-like cells that express collagen [52]. It still needs to be established if the presence of M2 macrophages identifies a profibrotic mechanism inherent to the pathogenesis of sarcoidosis rather than part of a generalized wound-healing response to lung inflammation and injury. AMs are also the main source, together with lung T cells, of interferon gamma (IFN*γ*), which is highly expressed in the BALF (Bronchoal Veolar Lavage Fluid) of sarcoidosis patients [53]. IFN*γ* inhibits the expression of macrophage peroxisome proliferator-activated receptor *γ* (PPAR*γ*), a negative regulator of inflammation. Under normal physiological conditions, macrophages constitutively express PPAR*γ*. PPAR*γ* promotes macrophage IL-10 production, which inhibits the release of TNF*α*, IL-12, and matrix metalloproteinases (MMPs) from dendritic cells (DCs). In sarcoidosis, PPAR*γ* activity is deficient in AMs [54], leading to an increase in the production of TNF*α*, IL-12, and MMPs, which cause lung damage and fibrosis and induction of T-cell chemotaxis [55]. Moreover, increased TNF*α* and decreased IL-10 expression liberate DCs from the inhibition by macrophages, initiating a self-amplifying inflammatory loop (see below)

Peripheral blood DCs from patients with sarcoidosis are less immunostimulatory than normal blood DCs and are less capable of mounting a delayed-type hypersensitivity (DTH) response [40]. Consistent with this finding, sarcoidosis lung DCs are less able to induce T cell proliferation than normal lung DCs [55]. A real paradox in sarcoidosis stands in the fact that, despite the partially blunted immunostimulatory capacity of circulating and pulmonary DCs, they are still effective inducers of T cell proliferation, more than macrophages themselves, both through direct TCR and through the secretion of potent inflammatory mediators, such as IL-12. Through IL-12, DCs are able to polarize Th1 T cells, stimulate T cell proliferation, and induce leukocyte chemotactic factors that contribute to granuloma formation [56]. Overall, the function of DCs in sarcoidosis is paradoxical. On the one hand, they are capable of initiating an antigen-driven, inflammatory oligoclonal T cell responses [57]. On the other hand, they are anergic and less immunostimulatory than normal DCs [40]. 

### 3.3. Adaptive Immune Cells

#### 3.3.1. T Cells

T-cells, particularly activated CD4+ T-cells, play a key role in the inflammation in sarcoidosis. Different subsets of CD4+ helper T cells participate in the immunopathogenesis of sarcoidosis. The main feature of the acute disease is represented by a Th1/Th17/regulatory T cells (Tregs)-driven inflammatory process involving macrophages both as antigen-presenting cells and key effectors. As discussed above, when triggered by factors as yet unidentified, APCs such as DCs release cytokines (e.g., IL-12) [55] and other inflammatory factors, leading to a milieu that induces recruitment and activation of Th1 CD4+ T-cells and monocytes to the lungs. In sarcoidosis, the lung homes up to ten times as many CD4+ T-cells as the peripheral blood, thus leading to an elevated CD4/CD8 ratio as measured in BAL fluid [58]. The CD4+ T cells that trigger the granuloma formation are strongly Th1 polarized. Upon TCR activation, the expression of IFN*γ* in CD4+ T cells becomes more pronounced. In response to IL-12, CD4+ T-cell production of IL-4 and IL-13 (cytokines that facilitate the fibroproliferative response) is inhibited. IL-12 and IL-18 act synergistically to promote the formation of sarcoid granulomas. As detailed above (see AM section) in sarcoidosis, the deficient PPAR*γ* activity in AMs [54] attracts more T cells and myeloid cells into the inflammatory milieu. CD4+ T-cell activation also increases IL-2 production, resulting in increased Th1 polarization. Interestingly, the persistent antigen exposure can induce anergy and/or exhaustion of the pathogenic CD4+ T cells, probably as a defense mechanism to temper down the chronic T cell activation and subsequent inflammation [59]. Pulmonary CD4+ T cells from patients with sarcoidosis spontaneously secrete IL-2 ex vivo, but upon TCR stimulation, they express less IL-2 and IFN-γ compared to CD4+ T cells from other lung diseases and healthy controls, in line with an anergic/exhausted phenotype [60]. Other signs of Th1 anergy/exhaustion due to prolonged antigen exposure in sarcoidosis are their reduced proliferative capacity and their higher levels of apoptosis, associated with increased programmed death (PD)-1 expression [61]. 

Th17 cells that express IL-17 and a specific master transcription factor, known as RORc [62], have a key role in the plasticity of granuloma formation and maintenance. They participate in the alveolitic/granuloma phase, the maintenance of granuloma, and the progression towards the fibrotic phase of sarcoidosis [63]. Th17 cells are increased in the lung and the peripheral blood of patients with active sarcoidosis. Their recruitment to granulomas is due to the release of cytokines and chemoattractants by locally and systemically activated macrophages. 

Failure of immune regulatory mechanisms to limit the duration of Th1–Th17 inflammation has been suggested to contribute to persisting granulomatous responses in sarcoidosis. T regulatory cells (Tregs) are vital for the suppression of cell-mediated immune responses. However, Tregs in the sarcoid granulomas (as opposed to peripheral Tregs) that have also been found elevated in BALF from patients with sarcoidosis undergo extensive amplification and are therefore impaired in their ability to repress immune responses [64]. On the other hand, they secrete pro-inflammatory cytokines (e.g., IL-4), which induces granuloma formation via mast cell activation and fibroblast amplification [65]. To date, the role of regulatory T cells (Tregs) in the pathogenesis of sarcoidosis remains controversial. Further studies are needed to shed light on the complex interactions between regulatory mechanisms and off-targeted immune responses in the disease. 

#### 3.3.2. B Cells

Little is known about B cell immunity in sarcoidosis [66]. Hypergammaglobulinemia (including autoantibodies) is well-recognized in sarcoidosis, where B cells form prominent infiltrates at the periphery of lung granulomas [67]. Thus far, whether B cells play a direct role in disease pathogenesis in sarcoidosis is unknown, but altered and/or increased (auto) antibody responses have been reported in these patients. Unlike healthy subjects, patients with sarcoidosis show a direct correlation between the number of BAL T cells and the proportion of BAL cells that secrete IgG [68]. Additionally, increased numbers of memory IgA-producing B cells are found in patients with sarcoidosis, suggesting that IgA could be involved in granuloma formation [69]. Interestingly, B cells might also play a role in the autoimmune phenomena associated with the onset and progression of sarcoidosis. In fact, several types of autoantibodies have been found, both circulating and associated with the lung tissue, in patients with sarcoidosis, such as anti-mitochondrial, anti-nuclear antibodies, and autoantibodies to double-stranded DNA [70]. Case-report evidence of responses to rituximab exists in extrapulmonary sarcoidosis but not for pulmonary disease as yet [71]. Recently, age-associated B cells (ABCs) [72], B cell subtypes associated with aging, in which they contribute to inflammation [73], have been found increased in peripheral blood and BAL (relative to healthy subjects) in patients with sarcoidosis [74] but their role needs to be clarified.

## 4. Pharmacological Treatment

In sarcoidosis, the decision to initiate treatment is not straightforward, mainly due to the highly variable and unpredictable course of the disease and the lack of evidence-based management guidelines. Broadly, the treatment is selected based on the presence of life- or organ-threatening disease and very poor quality of life. Figure 4 and Figure 5 summarize the approach to the management of sarcoidosis and the current therapeutic options, as seen by the authors. 

Systemic glucocorticoids (GCs) are the first-line therapy for pulmonary and extrapulmonary sarcoidosis, as they inhibit macrophage and lymphocyte activation and modulate several cytokines that participate in granulomatous inflammation. However, they do not alter the course of the disease, and their long-term use is associated with significant safety and tolerability issues. Therefore, the pros and cons of initiating GC treatment should be carefully weighed, bearing in mind that approximately half of the patients have self-remitting or non-progressive disease, which does not require therapy [75,76]. In patients with pulmonary sarcoidosis, treatment should be considered in the following circumstances: disabling and worsening respiratory symptoms (e.g., dyspnoea, cough, chest discomfort); severe functional impairment or progressive functional deterioration (e.g., TLC (Total Lung Capacity) decline of ≥10% and/or FVC (Forced Ventilatory Capacity) decline of ≥15% and/or DL_CO_ (Diffusion Capacity for Lung Carbon Monoxide) decline of ≥20% over three to six months); or major progression of radiographic abnormalities (e.g., worsening of interstitial opacities, development of cavities or progression of fibrosis with honeycombing) [77]. Some patients require treatment with GCs because of significant extrapulmonary manifestations such as ocular, cardiac, neurological and renal, and/or hypercalcemia. An additional potential indication for GC therapy includes severely impaired quality of life due to fever, fatigue, arthralgia, or disfiguring skin disease [1,78]. The optimal dose and duration of GC treatment are unknown. We generally initiate therapy with oral prednisone at a daily dose of 0.3–0.5 mg/kg of ideal body weight (usually 20 to 40 mg/day) depending on disease severity. We continue the initial dose for four to six weeks and then re-evaluate the patient. Provided clinical, functional, and radiographic features are stable or improved, we taper the dose by 5 to 10 mg every one to three months for a total treatment period of six to nine months [79,80]. 

The majority of patients are able to discontinue treatment after one year; however, relapse is common during tapering or after GC discontinuation, with as many as 30% of patients requiring longer-term therapy [81]. Overall, the GC use is associated with symptomatic and radiographic improvement, at least initially, but their long-term benefit is unclear [82]. The antimalarial drug hydroxychloroquine may be preferable to GC in patients with skin disease or hypercalcemia [83,84]. 

Immunosuppressants: for patients experiencing relapse/disease progression despite GC treatment or intolerable side effects of GCs, a step-up approach to second-line agents is recommended. Methotrexate (MTX) is the drug of choice for sarcoidosis selected in several consensus statements [85]. MTX is an antimetabolite with anti-inflammatory and immunosuppressive properties. Several case series and randomized trials suggest that MTX is safe and effective in sarcoidosis patients with lung, eye, skin, and central nervous system involvement [86]. A 2-year retrospective study of 200 patients showed that MTX and azathioprine, another immunosuppressant, has similar steroid-sparing capacity, beneficial effects on lung function, and safety and tolerability profiles, although the rate of infection was significantly higher in the azathioprine group [87]. MTX is generally given at an initial dose of 5–7.5 mg weekly, which is gradually increased weekly, either orally or intramuscularly. MTX has a slow onset of action, and its efficacy should be assessed after at least six months of treatment. Methotrexate is generally well tolerated [88], but its use may be associated with a variety of adverse effects, including liver and lung toxicity, increased risk of infection, and myelosuppression. In order to reduce the risks of myelosuppression, folic acid supplementation is given at a dose of 5 mg weekly, and tests to monitor blood cell counts and kidney and liver function are performed at regular intervals (i.e., every four to eight weeks). Other immunosuppressants used alternatively to MTX are mycophenolate mofetil (MMF), azathioprine, and leflunomide, although the evidence in support of their efficacy is less robust. In a retrospective study of patients with pulmonary sarcoidosis who either had not responded to a prior immunosuppressive agent or had experienced an adverse event necessitating discontinuation of the drug, addition of MMF allowed a reduction of prednisone dose but was not associated with improved lung function [89]. In a retrospective case series, patients with sarcoidosis received leflunomide with or without concomitant MTX for the pulmonary or ocular disease [86]. Complete or partial response to leflunomide was observed in the great majority of patients treated with leflunomide alone and in combination of leflunomide and MTX. Overall, leflunomide was well tolerated and appeared to be as effective as MTX but less toxic.

Biologicals: TNF-α antagonists are generally reserved for patients with active/progressive disease despite being treated with GCs and at least one second-line immunosuppressant agent, particularly in cases of organ- or life-threatening disease. Infliximab is a humanized monoclonal antibody that neutralizes TNF-α. In a phase II, randomized, double-blind, placebo-controlled study, 138 patients with chronic pulmonary sarcoidosis were randomized to receive i.v. infusions of infliximab or placebo at baseline and at weeks 2, 6, 12, 18, and 24 [90]. Infliximab use was associated with a mean significant increase of 2.5% from baseline to week 24 in the percentage of predicted forced vital capacity (FVC) vs. the placebo group. Post-hoc exploratory analyses suggested that the benefit of infliximab treatment might be greater in patients with more severe disease. Infliximab is safe and well-tolerated, and the proportions of patients who had adverse events are reported to be similar to those in the placebo groups. In a prospective, open-label trial, 56 patients with sarcoidosis refractory to conventional treatment received eight infusions of infliximab [91]. After 26 weeks of treatment, the infliximab group had a significant increase in FVC predicted vs. the placebo group and a reduction in the active disease in the lung parenchyma, as measured by ^18^F-FDG PET (Fluorodeoxyglucose Positron emission tomography) as maximum standardized uptake value (SUV_max_). Notably, change in lung function significantly correlated with the level of disease activity. Adalimumab is a fully human anti-TNF-α monoclonal antibody. In an open-label, single-center study, 11 patients with refractory pulmonary sarcoidosis received adalimumab for 45 weeks [92]. At the 24-week follow-up, FVC improved in four patients and stabilized in seven. In addition, a successful outcome—defined as a reduction in immunosuppressive therapy improvement in FVC of 5% or greater, improvement in 6 MWD (minute walking distance) of 50 m or greater—was observed in 9/11 patients (82%) and 8/10 patients (80%) at weeks 24 and 52, respectively. In addition, among 18 patients who switched from infliximab to adalimumab due to antibody formation or severe adverse events, seven (39%) experienced clinical improvement, six (33%) remained stable, and five (28%) deteriorated [93]. Severe adverse events occurred in four patients, including one who discontinued adalimumab due to a lupus-like reaction. Current evidence does not support the use of etanercept, which is a soluble TNF-α receptor, in patients with sarcoidosis. Indeed, a prospective, open-label, phase II trial of etanercept 25 mg twice weekly in patients with progressive pulmonary disease was terminated after the enrollment of 17 patients due to excessive clinical deterioration, development of intolerable side effects, and/or need for other immunosuppressive agents [94]. At present, there is no evidence supporting the routine use of TNF-α antagonists in sarcoidosis. However, they may be useful in selected cases of cardiac and neurological disease refractory to conventional treatment [95,96,97].

Besides TNF-α inhibitors, rituximab, a chimeric anti-CD20 monoclonal antibody that induces B cell depletion is the only biologic agent for which there is some evidence of efficacy. Sweiss and colleagues conducted a prospective, open-label, phase I/II trial of rituximab at baseline and two weeks later in patients with refractory pulmonary sarcoidosis (*n* = 10) [98]. At week 24, 5/10 patients had a >5% absolute improvement in FVC % predicted and 4/10 patients had a >10% improvement in FVC % predicted, whereas 2/8 patients had a >10% absolute improvement in FVC % predicted at week 52. Moreover, 6 min walking test (6MWT) improved from the initial value by >30 m in 5/10 patients and by >50 m in 3/10 patients at week 24 and by >50 m in 3/8 patients at week 52. Chest X-rays remained unchanged throughout the study. Whether rituximab may be a viable alternative to anti-TNF-α antibodies for refractory sarcoidosis is unclear and requires further studies.

Other novel therapies: several drugs targeting the immune system are currently being evaluated as future therapeutic strategies for sarcoidosis. The active inflammatory form of sarcoidosis is characterized by an exaggerated production of TNF-α and INF-γ. Fontolizumab, a humanized monoclonal antibody against INF-γ currently tested in Crohn disease [99], has the potential to depolarize M1 back to an inactive state. Other drugs target the M2 polarization that promotes sarcoidosis-related fibrotic processes in the lung. M2 polarization is regulated by STAT-6, a transcription factor that promotes a signaling cascade that includes the PPAR-γ pathway (see above). Treatment with Leflunomide, a tyrosine kinase inhibitor preventing STAT-6 phosphorylation that regulates inflammation by suppressing Th17 cells and promotes the function of Tregs, may improve lung function in patients with sarcoidosis [100]. Doxycycline and Dupilumab, two drugs that inhibit the M2 polarization, are also currently being tested in patients with sarcoidosis, but there is still lack of data enough to draw conclusions on their effectiveness. Other treatments, such as nicotine therapy, therapies targeting checkpoint inhibitors, and antibacterial therapy are also in the pipeline for the treatment of patients with sarcoidosis, but the available data are still scant [101].

Transplant: for highly selected patients with advanced fibrotic lung disease, severe pulmonary hypertension, or both, lung transplantation may be the only therapeutic option for prolonging survival and improving quality of life [102]. Bilateral lung transplantation is generally favored, as it appears to be associated with slightly better survival compared to single lung transplantation [103]. Notably, post-transplant survival is similar to that for other indications, such as idiopathic pulmonary fibrosis [102,104]. Heart transplantation has also been successfully performed in patients with cardiac sarcoidosis and end-stage heart failure [105]. Carefully selected patients with advanced heart failure due to cardiac sarcoidosis have an acceptable outcome after transplantation without evidence of recurrence of the disease in the allograft [106].

## 5. Conclusions

Sarcoidosis is a heterogeneous disease with diverse clinical presentations that likely mirror the wide umbrella of underlying immunological endotypes. Thus, a “one size fits all” approach to treating sarcoidosis is not applicable, as some patients may benefit from therapies targeting the excessive spread of the inflammatory responses, while others might benefit from therapies boosting their immune and/or regulatory responses. Further studies aimed at identifying the clinical phenotypes associated with specific blunted and/or exaggerated immune responses are needed in order to shed light on the possible therapeutic approaches for this multifaceted disease.

## Figures and Tables

**Figure 1 jcm-09-02363-f001:**
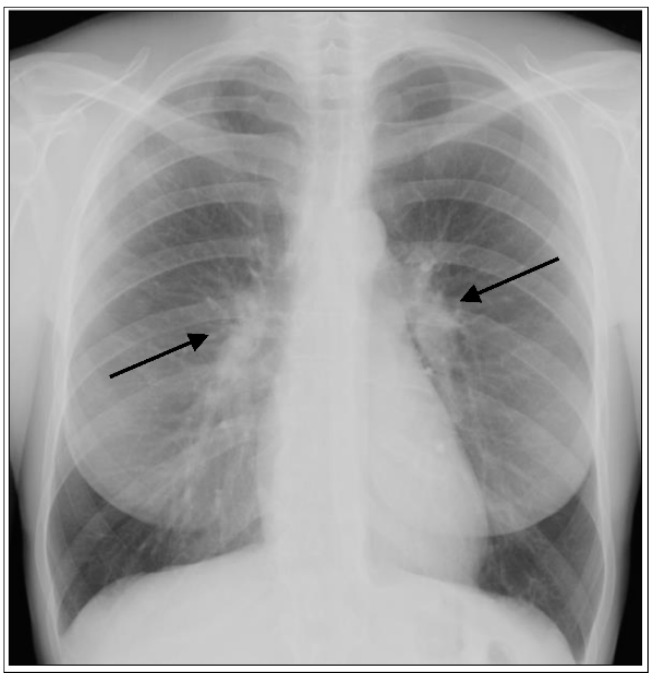
Acute form of sarcoidosis. Posterior-anterior chest radiograph of a 35-year-old female with Löfgren’s syndrome demonstrating bilateral hilar adenopathy (arrows) without evidence of parenchymal lung involvement.

**Figure 2 jcm-09-02363-f002:**
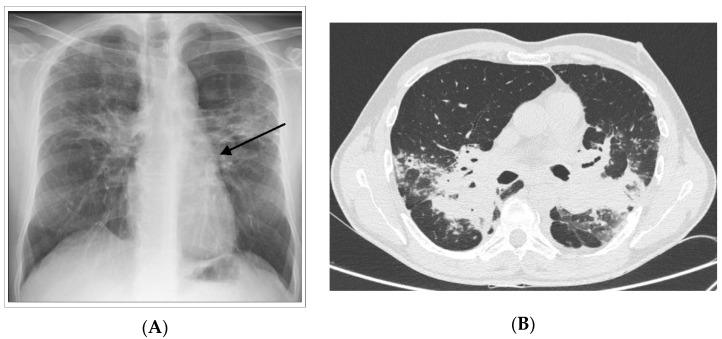
Chronic form of sarcoidosis. (**A**) Posterior-anterior chest radiograph of a 55-year-old man with progressive pulmonary fibrosis despite treatment demonstrating permanent, coarse linear opacities, radiating laterally from the hilum upward and outward (arrows). (**B**) high-resolution CT showing intra parenchymal enlargement of pulmonary arteries and enlarged pulmonary trunk at the bifurcation.

**Figure 3 jcm-09-02363-f003:**
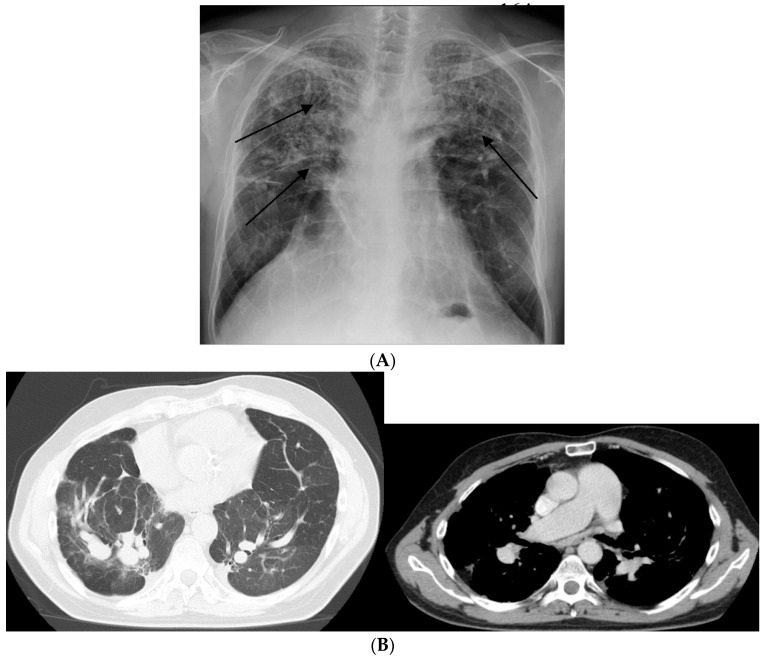
Chronic form of sarcoidosis complicated with pulmonary arterial hypertension (PAH). (**A**) Posterior-anterior chest radiograph of a 58-year-old man listed for lung transplant demonstrating extensive parenchymal scarring throughout both lungs, most marked in the upper lungs (arrows) and in the perihilar regions. (**B**) high-resolution CT showing intra-parenchymal enlargement of pulmonary arteries and enlarged pulmonary trunk at the bifurcation.

**Figure 4 jcm-09-02363-f004:**
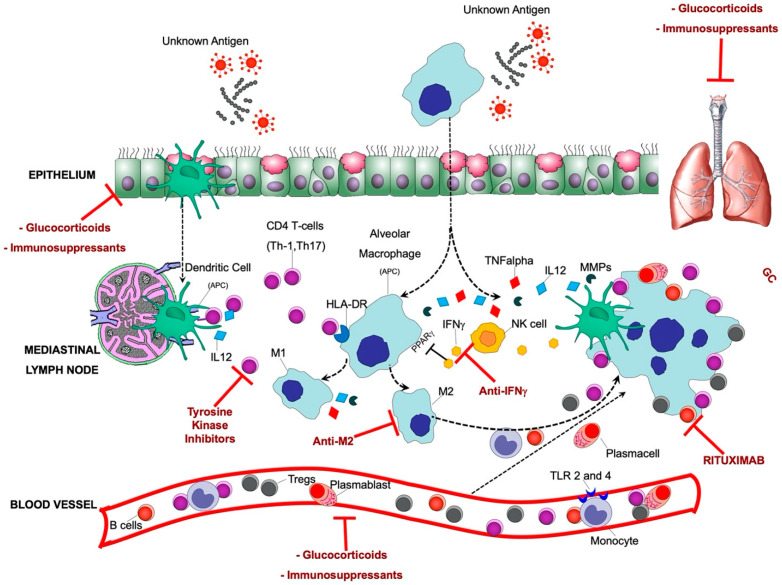
Pathobiology of sarcoidosis and current therapeutic approaches: An unknown airborne-antigen activates interstitial dendritic cells (DCs) and alveolar macrophages (AMs) simultaneously. The interstitial DCs pick up the antigen and migrate toward the mediastinal lymph nodes, where they serve as antigen-presenting cells (APCs) to T helper (Th)1 and 17 cells by initiating their differentiation and clonal expansion. Simultaneously, AMs: (1) serve as APCs together with DCs; (2) differentiate as M1 and M2 macrophages and further induce the inflammatory process (M1), fibroblasts proliferation, and fibrotic tissue deposition (M2); and (3) produce the pro-inflammatory cytokine tumor necrosis factor-α (TNF-α) via human leukocyte antigen (HLA)-DR, and other chemoattractants such as IL-12 under stimulation of both TNF-α and natural-killer (NK) cell-derived interferon-γ (INF-γ). Persistent stimulation, mediated by APCs leads to continuous recruitment of B cells, plasmacells, Th1/17 cells, monocytes (expressing higher Toll Like Receptor (TLR)2 and TLR4, inducing Th1 and Th2 responses, respectively), and regulatory T cells (Tregs) from the bloodstream to the lung, where they contribute to the granuloma formation. Tregs infiltrating the granuloma fail to diminish the exaggerated immune response, thereby contributing to granuloma persistence and integrity. The therapeutic targets that are currently being used/tested for sarcoidosis are also indicated.

**Figure 5 jcm-09-02363-f005:**
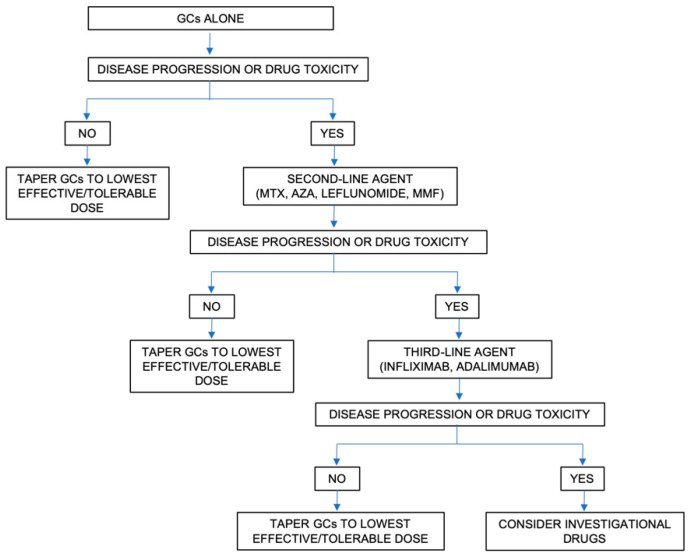
Approach to the therapeutic management of sarcoidosis. Abbreviations: AZA: Azathioprine; GCs: glucocorticoids; MMF: mycophenolate mofetil; MTX: methotrexate.

**Table 1 jcm-09-02363-t001:** Overall frequency of organ involvement at the time of diagnosis (patients could have more than one organ involved) adapted from Baughman RP et al., 2001. [4].

Organ Involvement	Frequency, %
Lung	90
Skin (excluding EN)	16
Erythema nodosum (EN)	8
Eye	12
Extrathoracic lymph node	15.2
Liver	12
Spleen	7
Neurologic	5
Cardiac	2

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
