# Peer review of "Clinical Presentations, Pathogenesis, and Therapy of Sarcoidosis: State of the Art"

_jcm, 2020, doi:10.3390/jcm9082363_

Round 1

Reviewer 1 Report

MAJOR COMMENTS

Overall a nice review of sarcoid

Cardiac sarcoid (CS) is recognised as an increasingly important type of disease due to morbdity and mortality in otherwise well patients. Can the authors consider separating it from the 'brain and heart' section? And can they explain that the heart can be involved in many ways: true CS due to granulomas, pulmonary hyertension due to lung disease, and increased risk of ischaemic heart disease in sarcoid.

T-cell section.  They mention antigens but would be good if they can go into more detail. There are a number of recent papers by various groups looking at mycobacterial antigens (eg W Drake's papers in the last decade) and also self antigens such as vimentin (eg Grunewald and/or eberhardt et al). These give insights into the possible autoimmunity in sarcoid and should be mentioned

Fibrosis. They mention the risks of lung fibrosis in sarcoid. This would be a good chance to discuss antifibrotics in progressive lung fibrosis (inlcuding data from last year's NEJM nintedanib paper and should these drugs be given?)

Transplant. It would be nice to have a few sentences on transplant (lung vs heart/lung) in sarcoid.

MINOR COMMENTS

Abstract is missing a word (in): Sarcoidosis is a systemic disease of unknown etiology characterized by the presence of 10 noncaseating granulomas that can occur IN any organ

In the 'brain' section at the beginning can they make it clear that central nervous involvement is different to peripheral nerves?

Steroids: can they give a clearer idea (in their own views as there is no good evidence) about how they drop steroid dose ie down to what level over 3 months, how to taper off completely. It would also be good to explain that patients should be at the heart of these decisions.

Can they go into more detail about risks of immunsupression and need for blood monitoring, use of folic acid when giving MTX etc 

There is no good evidence for TNFa and similar drugs. They should make this clearer but also say that sometimes these drugs are targeted for specific forms of sarcoid eg active cardiac or neuro diseases

Author Response

Please find the responses attached. Thanks

Reviewer 2 Report

This review on the clinical manifestations, pathophysiology, and the pharmacologic treatment of sarcoidosis is comprehensive and well-written.

There are, however, some minor comments that the authors need to address or discuss:

  1. Lines 76–78: The authors should provide some references in support for the proportions provided for Lofgren’s syndrome’s prognosis.
  2. Line 85: Anterior uveitis is a relative uncommon extra-pulmonary manifestation of Lofgren’s syndrome. The classic Lofgren’s triad is bihilar lymphadenopathy, erythema nodosum and periarticular inflammation.
  3. Lines 122–123: Although commonly used, the “5%” case fatality (proportion) is difficult to interpret in the absence of time scale. With regards to mortality, it would be more interesting to see if death happens earlier in sarcoidosis patients compared to what is expected in the general population. In addition, in individuals who die from sarcoidosis, cardiovascular disease, infections, and cancers are more common underlying causes of death than pulmonary fibrosis (see https://doi.org/10.1183/13993003.00457-2016 and https://doi.org/10.1183/13993003.01815-2017).
  4. In the “Pathogenesis of sarcoidosis” section, I think the reader would appreciate a small paragraph on potential antigens that are though to trigger the sarcoid inflammatory processes. Otherwise, this section is very comprehensive and well-written.
  5. Lines 337­­­–338: A reference is needed for the statement on hydroxychloroquine.
  6. Title: The authors should consider changing “therapy of sarcoidosis” to “pharmacologic treatment of sarcoidosis”.

Author Response

Please find the responses attached. 
